# Potential Chemopreventive Effects of Dietary Combination of Phytochemicals against Cancer Development

**DOI:** 10.3390/ph16111591

**Published:** 2023-11-10

**Authors:** Takuji Tanaka, Ryogo Aoki, Masaru Terasaki

**Affiliations:** 1Department of Diagnostic Pathology, Gifu Municipal Hospital, 7-1 Kashima-cho, Gifu 500-8513, Japan; raoki@gmhosp.gifu.gifu.jp; 2School of Pharmaceutical Sciences, Health Sciences University of Hokkaido, 1757 Kanazawa, Ishikari-Tobetsu, Hokkaido 061-0293, Japan; terasaki@hoku-iryo-u.ac.jp; 3Advanced Research Promotion Center, Health Sciences University of Hokkaido, 1757 Kanazawa, Ishikari-Tobetsu, Hokkaido 061-0293, Japan

**Keywords:** chemoprevention, combination, phytochemical, natural and synthetic chemicals, oncogenesis

## Abstract

Cancer remains a major cause of cancer-related death worldwide. Over 70% of epithelial malignancies are sporadic and are related to lifestyle. Epidemiological studies suggest an inverse correlation between cancer incidence and fruit and vegetable intake. Numerous preclinical studies using in vitro (cell lines) and in vivo animal models of oncogenesis have reported the chemopreventive effects of dietary phytochemical agents through alterations in different biomarkers and signaling pathways. However, there is contrasting evidence from preclinical studies and clinical trials. To date, the most studied compounds include curcumin, resveratrol, isoflavones, green tea extract (epigallocatechin gallate), black raspberry powder (anthocyanins and ellagitannins), bilberry extract (anthocyanins), ginger extract (gingerol derivatives), and pomegranate extract (ellagitannins and ellagic acid). Overall, the clinical evidence of the preventive effects of dietary phytochemicals against cancer development is still weak, and the amount of these phytochemicals needed to exert chemopreventive effects largely exceeds the common dietary doses. Therefore, we propose a combination treatment of natural compounds that are used clinically for another purpose in order to obtain excess inhibitory efficacy via low-dose administration and discuss the possible reasons behind the gap between preclinical research and clinical trials.

## 1. Introduction

Cancer is a leading cause of death and an important barrier to prolonging life worldwide. As estimated by Parkin [1] in 2000, approximately 19.3 million new cancer cases and almost 10.0 million cancer deaths occurred in the United States in 2020 [2,3]. Because of the heterogeneous characteristics of malignancy, which are associated with irreversible impairment of cellular homeostasis and function, perfect therapy is difficult to achieve. Potential cancer treatment strategies include chemotherapy, surgery, and radiotherapy. Conventional therapies for cancer pose many challenges, including toxicity, multidrug resistance, and high economic costs. Despite significant advances in the diagnosis and treatment of cancer, it remains a fatal disease because of the lack of prevention, difficulty associated with making an early diagnosis, and lack of effective drugs. Radiotherapy, chemotherapy, and surgery are not only expensive, but also produce a number of side effects that are detrimental to the patient’s quality of life. Therefore, there is an urgent need to discover cancer cell-specific anticancer therapies that are affordable, safe, and well-tolerated by patients. In this context, technical advances in genome sequencing and the implementation of next-generation sequencing (NGS) in clinical oncology have enabled us to present a method for individualizing cancer patient therapy based on molecular profiles [4,5]. However, the frequency of cancer patients with druggable mutations is relatively low (approximately 10% of patients undergo genomic analysis by NGS) [6].

Carcinogenesis is a complex multistep process initiated by exposure to genotoxic or carcinogenic agents. In the initial stage of carcinogenesis, a single normal cell undergoes DNA damage and transforms into a cancerous cell. The initiated cells begin abnormal multiplication, resulting in a heterozygous tumor cell population. In the tumor promotion stage, actively proliferating preneoplastic cells accumulate and then reach the tumor progression stage, where tumor cells have malignant potential for invasion and metastasis. Cancer chemoprevention, a term proposed by Dr. Sporn [7] in 1976, is defined as the application of natural or synthetic agents to inhibit, reverse, or hinder the tumorigenesis/carcinogenesis/oncogenesis process. Numerous studies have reported that phytochemicals act as chemopreventive agents by interfering with specific regulatory stages of carcinogenesis [8,9]. The overproduction of oxidants causes oxidative damage to DNA, proteins, and lipids, which aids in cancer pathogenesis [10]. Dietary phytochemicals with potential antioxidant properties that reduce or prevent oxidative stress can serve as chemopreventive agents [11,12]. A close association between cancer and inflammation is evident because chronic inflammatory stimuli contribute to oncogenesis [13,14,15,16]. Phytochemicals with anti-inflammatory properties may have potential chemopreventive properties [17,18,19,20,21,22,23]. In addition, several phytochemicals regulate the proliferation and differentiation of cancer cells, cause cell cycle arrest at different phases, and regulate cell death pathways such as apoptosis and autophagy [24,25,26]. Professor Wattenberg proposed a conventional classification of chemopreventive agents, namely blocking and suppressing agents [27,28,29]. The former acts during the initiation stage and the latter during the promotion/progression stage(s). Generally, chemopreventive agents are used as single treatments. However, as proposed by Dr. Sporn [30], combined treatment with different chemopreventive agents can be more effective than a single treatment, with fewer side effects or toxicity [31,32,33].

In this short review, we briefly summarize recent advances in combination chemoprevention of phytochemicals and drugs that have cancer chemoprevention effects. 

## 2. Human Carcinogens

Chemical carcinogens are agents that cause cancer in both humans and animals. Chemical carcinogens may be involved in all stages of carcinogenesis. They are also divided into two categories based on the pathogenic mechanisms of carcinogenesis [34]: non-genotoxic and genotoxic agents. They constitute 10–20% of all carcinogens [35]. Other chemical carcinogens can induce cancer through mechanisms other than genotoxic or non-genotoxic mechanisms. Genotoxic carcinogens cause DNA damage either directly or after metabolic activation owing to their electrophilic nature [36]. Electrophiles are electron-seeking molecules that form adducts with intracellular nucleophilic macromolecules [37,38,39]. When repair mechanisms prior to replication fail to repair the damage, DNA adduct formation may result in cancer development [37,38,39]. 

As exogenous chemical exposure is associated with the production of reactive electrophilic species (RES), there is an association between RES and DNA adduct formation, mutations, and finally cancer [40]. Importantly, direct-acting (or activation-independent) carcinogens interact directly with DNA or cellular components owning electrophilic groups [41], whereas activation-dependent (or indirect-acting) carcinogens require enzymatic activation to electrophilic forms in order to become carcinogenic or reactive intermediates that exert genotoxic effects [42,43]. The harmful properties of carcinogenic agents are mediated by exogenously or metabolically generated electrophiles and reactive oxygen species (ROS) [44,45].

A free radical is an independent molecular species containing an unpaired electron. Many radicals are unstable and highly reactive [46]. The balance between various oxidants maintains cellular homeostasis and protects cells against oxidative stress [46]. Excessive quantities of oxidants contribute to cellular damage [47] because of their ability to oxidize lipids, which are essential components of cell membranes and protein products. DNA adducts are formed when ROS react with DNA [48,49]. DNA is considered the main target of oxidative damage, especially in aging and cancer. The formation of free radicals and other ROS is mediated via metabolic processes or external exposure to air pollutants, cigarette smoke, radiation, heavy metals [48,49], industrial solvents, certain drugs, pesticides, and other xenobiotics [46,50]. Environmental stressors and xenobiotics contribute to increased ROS production [46,50]. An imbalance between ROS and the antioxidant defense system has been implicated in all stages of carcinogenesis [26]. ROS are also involved in the resistance to therapy. Additionally, ROS may alter the genes associated with apoptosis, proliferation, and transcription factors [51]. However, enzymatic antioxidants, such as superoxide dismutase (SOD), catalase, glutathione system enzymes (glutathione reductase, glutathione peroxidase, glutathione *S*-transferase), and non-enzymatic antioxidants (vitamin E, vitamin C, glutathione, and numerous dietary phytochemicals) are able to protect biomolecules either directly or indirectly from oxidative damage [52,53]. 

## 3. Metabolic Activation of Carcinogens

Most environmental carcinogens exist in the form of procarcinogens, and thus require metabolic activation to exert their carcinogenic ability. However, metabolic processes may also cause inactivation, detoxification, and an increase in the aqueous solubility of compounds, resulting in detoxification and excretion from the body. The activation of carcinogens through various metabolic processes usually generates electrophilic reactive intermediates with the ability to bind to DNA, form DNA adducts, and finally contribute to mutations [36,42], which are the events that initiate oncogenesis [54]. 

The bioactivation of carcinogens into reactive electrophiles that are capable of covalently binding to DNA is mediated primarily through xenobiotic-metabolizing enzymes, mainly cytochrome P450s, also known as CYPs [36,55]. However, other enzyme systems are involved in the activation of various carcinogens [42,56]. CYPs are defined as enzymes that function as major oxidative catalysts, which metabolize xenobiotic and endogenous compounds and activate carcinogens independently or in conjugation with phase II enzymes [36,42]. The major human CYP enzymes involved in the activation of chemical carcinogens are 1A1, 1A2, 1B1, 2A6, 2A13, 2E1, and 3A4 [57]. Subsequently, reactive metabolites bind to DNA and generate DNA adducts that, if not repaired, lead to damage and mutation in genes and cancer. Important environmental carcinogens, such as polycyclic aromatic hydrocarbons (PAHs), heterocyclic amines (HAAs), and tobacco-related nitrosamines, need to be activated through xenobiotic-metabolizing enzymes to initiate cell transformation [58,59,60,61]. HAAs and PAHs can initiate epithelial cell differentiation in various tissues. Their carcinogenicity is related to their interactions with the aryl hydrocarbon receptor (AhR) [62,63]. The CYP1 family is known to increase their transcription [62,63]. Most HAAs are mutagenic and carcinogenic, suggesting that HAAs contribute to the etiology of human malignancies related to dietary intake, as they are present in processed protein-rich foods such as meat [7], especially when cooked at a high temperature [58,59,60,61]. Among the HAAs, 2-amino-1-methyl-6-phenylimidazo [4,5-*b*]pyridine (PhIP) is classified as possibly carcinogenic to humans by the International Agency for Research on Cancer (IARC) [58,59,60,61], and is considered to be one of the most abundant HAAs formed in meat prepared at high temperatures and in tobacco smoke [58,59,60,61].

Metabolically activated tobacco-specific nitrosamines, 4-(methylnitrosamino)-1-(3-pyridyl)-1-butanone (NNK) and N′-nitrosonornicotine (NNN), which are classified by the IARC as carcinogenic to humans, also contribute to the formation of DNA adducts and malignancy [64,65]. The metabolite of NNK, 4-(Methylnitrosamino)-1-(3-pyridyl)-1-butanol (NNAL), is considered to be a strong carcinogen, and is found in the urine of both smokers and non-smokers exposed to second hand smoke [64,65].

## 4. Phytochemicals

Phytochemicals, which are defined as non-nutrient plant secondary metabolites, are present in fruits, vegetables, and grains. They have been reported to reduce the risk of various diseases, including malignancies [12,17,26,66,67,68]. The most studied phytochemicals are terpenoids, alkaloids, isothiocyanates, and polyphenols [21]. Phytochemicals function through various overlapping and complementary mechanisms, such as (i) antioxidants, (ii) detoxifying abilities, (iii) binding/dilution of carcinogens in the digestive tract, (iv) epigenetic alterations, and/or (v) modulation of cellular and signaling pathways [22]. Certain phytochemicals exert strong antioxidant and free radical scavenging activities, prevent DNA damage, and consequently inhibit the initiation of carcinogenesis [69]. Phytochemicals can promote detoxification and enhance the excretion of exogenous/endogenous carcinogens [70] by inhibiting Phase I enzymes that bioactivate carcinogens or induce Phase II enzymes [36]. Chronic proinflammatory signaling caused by a continuous imbalance in redox homeostasis is known to produce pro-oncogenes or anti-apoptotic factors [71]. Phytochemicals are potent modulators of such proinflammatory/inflammatory signaling pathways, which are activated by major transcription factors, nuclear factor kappa B (NF-κB), the signal transducer and activator of transcription 3 (STAT-3), and cyclooxygenase (COX)-2 [72]. Additionally, phytochemicals inhibit proliferation and induce the apoptosis of preneoplastic and neoplastic cells during oncogenesis [17,73]. Interestingly, environmental-exposure-related epigenetic impairment may cause damage to the fetus, thus influencing the disease risk later in life [74]. However, phytochemicals can reverse the epigenetic changes that occur during carcinogenesis [75]. Moreover, bioactive food compounds that modulate epigenetic markers reduce inflammatory responses by suppressing NF-κB activation [76]. Therefore, bioactive foods may initiate protection against epigenetic modifications throughout life. Several experimental studies have shown that phytochemicals modulate the formation of carcinogens or protect cells against carcinogen exposure [77].

Representative phytochemicals that exert their cancer-preventive effects on oncogenesis in preclinical and/or clinical studies are curcumin, epigallocatechin gallate (EGCG), ginsenoside Rg3, β-carotene, lycopene, sulforaphane, and resveratrol [12,78].

## 5. Combined Chemoprevention Strategy

Cancer chemoprevention is a promising strategy for blocking, reversing, or retarding carcinogenesis. Chemoprevention by administration of a single agent has been well studied, and a number of experimental reports have been published. However, the number of clinical studies is relatively small. The combination regimens include natural compounds (phytochemicals), synthetic chemicals (D,L-α-difluoromethylornithine (DFMO), non-steroidal anti-inflammatory drugs (NSAIDs), statins, and others), and natural compounds and NSAIDs. Finally, a natural compound may be used to reduce the toxicity of anticancer drugs or radiation. Phytochemicals have been reported to be rational, safe, non-toxic, and biologically active. Thus, the administration of phytochemicals is a promising way to prevent cancer development, especially in high-risk populations [69,79]. Vegetables, fruits, nuts, soy, tea, spices, whole grains, and edible macro-fungi, which have a variety of beneficial health effects, are rich sources of phytochemicals [8,9]. Numerous epidemiological and experimental studies have demonstrated that phytochemicals are essential for the prevention and management of epithelial malignancies [23].

Phytochemicals have protective effects against oncogenesis through various mechanisms, including inhibition of cell proliferation, induction of cell apoptosis and autophagy, suppression of cell invasion and migration, anti-angiogenesis, and regulation of the microenvironment [66]. In addition, certain phytochemicals can enhance the sensitivity of malignant cells to chemotherapeutic drugs [80,81,82].

In this regard, we show examples that prove the efficacy of combination treatment with diosmin and hesperidin, both of which have anti-inflammatory properties [18,83,84,85,86,87,88], against chemically induced carcinogenesis in several rodent tissues [18,84,89,90,91,92,93,94,95,96,97,98].

### 5.1. The Natural Sources of Diosmin and Hesperidin

A natural flavone glycoside with a molecular weight of 608.549 g/mol, diosmin (diosmetin 7-O-rutinoside, C_28_H_32_O_15_, Figure 1a), commonly presents in citrus plants belonging to the rutaceae family such as tangerine (*Citrus reticulata*). Diosmin is also made via the oxidation of hesperidin (Figure 1b), which is a corresponding flavanone glycoside, present in the pericarp of various citrus fruits. Citrus fruits such as lemon, sweet orange (*Citrus sinensis*), and grapefruits are rich sources of hesperidin (C_28_H_34_O_15_, Figure 1b), which is known as a flavanone glycoside. Hesperidin is also present in unripe sour oranges, Ponderosa lemon, *Citrus unshiu*, and *C. mitis*. The limited availabilities of both compounds are known. However, inclusion complexes of diosmin and hesperidin with cyclodextrins, such as β-cyclodextrin and 2-hydroxypropyl-β-cyclodextrin, enhance its solubility.

### 5.2. Combination Treatment with Two Natural Compounds, Diosmin and Hesperidin

Combinations of phytochemicals can promote cell death, suppress cell proliferation and invasion, sensitize cancer cells, and enhance the immune system, thus making them attractive alternatives for cancer treatment. Despite the vast number of rodent and preclinical cancer studies using individual phytochemicals, few research studies and clinical trials using phytochemical combinations in cancer treatment have been conducted to determine the combined effects of these compounds when used in synergistic, additive, or antagonistic combinations. We previously investigated the effect of two phytochemicals (diosmin and hesperidin) at their bioavailable levels on chemically induced carcinogenesis in rodents [89,90,91,99]. These studies showed that diosmin and hesperidin synergistically inhibited cell proliferation and induced apoptosis. In addition, the chemopreventive ability of citrus pulp and juices containing high amounts of β-cryptoxanthin and hesperidin was investigated using animal carcinogenesis models of the colon, tongue, and lungs [18,23,100,101]. In addition, citrus pulp and juices containing high amounts of β-cryptoxanthin and hesperidin exert cancer chemopreventive abilities and increase detoxifying enzymes in several tissues [18,101].

### 5.3. The Effects of a Single Administration of Diosmin, Hesperidin, or the Combination of Both Compounds

The animal models used include *N*-butyl-*N*-(4-hydroxybutyl)nitrosamine (OH-BBN)-induced mouse urinary bladder carcinogenesis [91], azoxymethane (AOM)-induced rat colon carcinogenesis [89], *N*-methyl-*N*-amylnitrosamine (MNAN)-induced rat esophageal tumorigenesis [99], and 4-nitroquinoline 1-oxide (4-NQO)-induced oral carcinogenesis [90].

Animals received 0.1% diosmin, 0.1% hesperidin, and 0.09% diosmin + 0.01% hesperidin in their diet during the initiation and post-initiation phases [91]. In the OH-BBN-induced mouse urinary bladder carcinogenesis model, dietary feeding with 0.1% diosmin, 0.1% hesperidin, and 0.09% diosmin + 0.01% hesperidin during the initiation phase caused a 66%, 79%, and 52% reduction, respectively, compared to the incidence rate of bladder urothelial carcinoma in mice that received OH-BBN alone (62%) [91]. When fed these diets during the post-initiation phase, the reduction rates were 87% with a 0.1% diosmin-containing diet, 68% with a 0.1% hesperidin-containing diet, and 66% with a 0.09% diosmin + 0.01% hesperidin-containing diet [91]. These reductions were all statistically significant (*p* < 0.05) in comparison to the carcinogen-alone treatment group [91].

In the AOM-induced rat colon oncogenesis model, dietary feeding with 0.1% diosmin, 0.1% hesperidin, and 0.09% diosmin + 0.01% hesperidin during the initiation phase caused a 70%, 93%, and 73% reduction, respectively, compared to incidence of colorectal adenocarcinoma in rats that received AOM alone (71%) [89]. When the animals were fed these diets during the promotion phase, the reduction rates were 93% with a diet containing 0.1% diosmin, 79% with a diet containing 0.1% hesperidin, and 93% with a 0.09% diosmin + 0.01% hesperidin-containing diet [89]. These inhibition rates were all statistically significant (*p* < 0.05) compared to the carcinogen-alone treatment group [89].

In the MNAN-induced rat esophageal tumorigenesis model, dietary feeding with 0.1% diosmin, 0.1% hesperidin, and 0.09% diosmin + 0.01% hesperidin during the initiation phase induced a 61%, 40%, and 20% reduction, respectively, compared to the incidence of colorectal adenocarcinoma in rats that received MNAN alone (75%) [99]. When animals were fed these diets during the promotion phase, the reduction rates were 48% with a 0.1% diosmin-containing diet, 25% with a 0.1% hesperidin-containing diet, and 5% with a 0.09% diosmin + 0.01% hesperidin-containing diet [99]. The inhibition rate after feeding with 0.1% diosmin was statistically significant (*p* < 0.05) compared to that in the carcinogen-alone treatment group [99].

In the 4-NQO-induced rat tongue carcinogenesis model, dietary feeding with 0.1% diosmin, 0.1% hesperidin, and 0.09% diosmin + 0.01% hesperidin during the initiation phase induced 68%, 75%, and 69% reductions, respectively, compared with the incidence of tongue squamous cell carcinoma in rats that received 4-NQO alone (65%) [90]. When fed these diets during the post-initiation phase, the reduction rates were 77% with a 0.1% diosmin-containing diet, 62% with a 0.1% hesperidin-containing diet, and 77% with a 0.09% diosmin + 0.01% hesperidin-containing diet [90]. These inhibition rates were all statistically significant (*p* < 0.05) compared to the carcinogen-alone treatment group [90]. Dietary administration of diosmin, hesperidin, and diosmin + hesperidin lowered the proliferative activity of the lesion and surrounding normal mucosa in target tissues [90]. These treatments also lowered tissue and/or blood polyamine levels, suggesting that polyamines are related to oncogenesis, and inhibition of polyamine biosynthesis may result in inhibition of carcinogenesis [89,90,99,102].

To determine the effects of combined treatment with 900 ppm diosmin and 100 ppm hesperidin on chemically induced urinary bladder, colon, esophageal, and tongue carcinogenesis in rodents, the dosages of these compounds were the same as those of daflon, which is clinically used in Europe to improve venous insufficiency in patients who undergo mastectomy for the treatment of breast cancer [103,104,105,106]. Single treatment with diosmin (1000 ppm) or hesperidin (1000 ppm) and combined treatment with both compounds during the initiation and post-initiation phases effectively suppressed cancer development in the urinary bladder [91], colon [89], and tongue [90]. However, single and combined treatments with test compounds during the initiation stage did not significantly inhibit esophageal carcinogenesis, although treatments during the post-initiation phase suppressed esophageal cancer development [99]. These findings suggest that the effects of the combination of diosmin and hesperidin differ among the phases of carcinogenesis in certain tissues. However, clinical application of this combination is possible because of its reduced toxicity [25,107,108]. In humans, the initiation and promotion of carcinogenesis is thought to continue after birth. Therefore, the effects of this combination on preneoplastic lesions in several tissues must be considered in the future.

When considering the mechanisms of action of the test compounds diosmin and hesperidin, lowering cellular proliferation in the target tissues is important because cell proliferation contributes to the growth of preneoplasia, although we did not clarify the molecular mechanisms of the inhibition/suppression ability of the compounds in the organs.

### 5.4. The Molecular Targets of Diosmin and Hesperidin

Several molecular targets of both compounds have been reported by several researchers. They include molecules involved in inflammation, the signal transducer of activators of transcription (STAT) pathway, oxidative stress, apoptosis, cell cycle, the phosphatidylinositol-3-kinase (PI3K)/protein kinase B (AKT) pathway, angiogenesis, activating protein-1, the extracellular signal-regulated kinase (ERK)1/2 mitogen-activated protein kinase (MAPK) pathway, and drug-metabolizing enzymes [109,110,111,112].

### 5.5. Combination Treatment with a Natural Compound and Synthetic Chemical (Drug)

Cancer is a heterogeneous disease with a wide variety of etiologies. Therefore, conventional monotherapies, including chemotherapy, have limited efficacy [113]. Additionally, several anticancer drugs exert undesirable adverse effects such as cardiotoxicity caused by doxorubicin [114], ototoxicity as a long-term side effect of cisplatin [115], and cognitive impairment by 5-fluorouracil [116]. Severe toxicities, including colitis, digestive perforation, toxic cardiomyopathy, pneumonitis/interstitial lung disease, acute respiratory distress syndrome, posterior reversible encephalopathy syndrome, necrotizing fasciitis, acute renal failure, and hypersensitivity, have been observed in patients who received molecular-targeted therapies such as antiangiogenic agents, anti-epidermal growth factor receptor (EGFR) therapy, and anti-human epidermal growth factor receptor type 2 (HER2) therapy [117]. Several plant-derived natural compounds, known as phytochemicals, are used together with drugs to enhance drug sensitivity. The investigation of phytochemicals has also proven to be a potential approach to discover new, effective, and safer anticancer agents [118]. Moreover, phytochemicals can inhibit cancer development by inducing apoptosis, modulating the immune response, suppressing angiogenic factors, and targeting gene expression in cancer [119]. In preclinical studies, natural products combined with chemotherapy have been shown to enhance anticancer activity and overcome drug resistance [120]. Moreover, a single high-dose administration of a natural compound may not be as effective as the administration of lower doses in combination anticancer treatment models [113]. The advantage of using a combination approach in cancer therapy is that it targets different pathways in a distinctive, synergistic, or additive manner [121]. In this context, the expected cross-resistance and overlapping adverse effects of these compounds should be considered when designing an experimental combination model.

### 5.6. Combination Treatment with Synthetic Chemicals (Drugs) with Chemopreventive Effects

Cancer remains a significant global health problem. A deep understanding of cancer biology, the underlying mechanisms of cancer development, and the identification of specific molecular targets for therapy/prevention may facilitate the development of novel therapeutic options. Drug repurposing is attractive because it has several advantages, including cost-effectiveness and better safety, compared to the development of new drugs and compounds.

NSAIDs and selective cyclooxygenase (COX)-2 inhibitors (coxibs) have been demonstrated to be promising and attractive candidates for clinical chemoprevention strategies for colorectal cancer (CRC), as supported by a large number of rodent and epidemiological studies that have clearly demonstrated that NSAID consumption prevents adenoma formation and decreases the incidence of CRC and its associated mortality. Aspirin chemoprevention may be effective in preventing CRC in the general population, whereas aspirin and celecoxib may be effective in preventing adenomas in patients after polypectomy. However, the consumption of NSAID and COX-2 inhibitors may result in serious adverse events in the gastrointestinal, renal, and cardiovascular systems. Therefore, studies on the use of lower doses in combination with other chemopreventive agents are warranted.

Metformin and aspirin have been explored as emerging chemopreventive agents for different types of cancers. For metformin, the most important mechanism may involve the inhibition of mammalian target of rapamycin (mTOR) signaling via the AMP-activated protein kinase (AMPK)-dependent and AMPK-independent pathways. The major mechanism of aspirin is its anti-inflammatory action through the inhibition of COX-1/COX-2 and modulation of the nuclear factor kappa B (NF-κB) or signal transducers and activators of transcription 3 (STAT3) pathway. Aspirin also activated AMPK. Both drugs may affect Notch, Wnt/β-catenin, and other signaling pathways. Therefore, the combination of metformin and aspirin may provide additive and synergistic effects for the prevention and treatment of cancer in different tissues.

Statins are lipid-lowering drugs used by large populations owing to their safety and ability to reduce cardiovascular disease and death [122]. In addition to their lipid-lowering abilities, statins also have various other effects, such as modulation of cell growth, suppression of inflammation, and induction of apoptosis [123]. Statins also have potential chemopreventive properties against CRC [124]. Statins tested their potential chemopreventive ability against cancer include pitavastatin, pravastatin, Fluvastatin, simvastatin, lovastatin, and atorvastatin. Experimental evidence suggests the efficacy of statins in inhibiting the growth of CRC in both in vitro human CRC cells and in vivo animal studies with xenografts, genetically predisposed animal models, and carcinogen-induced CRC models, individually or in combination with coxibs [125]. Although previous small prospective studies reported weak or no significant reduction in the risk of CRC with statin use at any dose level [126], the Molecular Epidemiology of Colorectal Cancer study reported a 45% reduction in the risk of CRC with five years of statin use, and a retrospective cohort study reported a 35% reduction in CRC risk in US veterans [127].

Endogenous and exogenous sources of polyamines have a significant impact on tumor growth. An irreversible inhibitor of ornithine decarboxylase, the first enzyme in the synthesis of polyamines, D,L-α-difluoromethylornithine (DFMO), has been shown to be an effective chemopreventive agent for various cancers [128,129,130]. Polyamines are reported to protect tumors from host immune responses by acting as natural immune suppressors of natural killer (NK) cell function. Polyamine deprivation stimulates NK cell activity. Although statins have been less investigated for their immune-modulating capabilities in CRC development, a low-dose combination of DFMO and rosuvastatin exerted additive chemopreventive efficacy and significantly enhanced innate immune cell function, including NK cells in colonic tumors, compared to individual dosing of DFMO and rosuvastatin [131].

Combination drugs include DFMO + selecoxib or sulindac [132], metformin + aspirin [133], selecoxib + AEE788 (EGFR tyrosine kinase inhibitor) [134], slindac + DFMO [135,136,137], piroxicam + DFMO [138,139], and selecoxib + lovastatin [140,141].

## 6. Cellular Senescence and Chemoprevention

Since the first report of senescence in the 1960s, when Hayflick and Moorhead [142] found that human fibroblasts had a limited ability to proliferate in culture, cellular senescence has been implicated in several physiological processes including aging, wound healing, development, and cancer prevention [143]. “Cellular senescence” refers to a cellular state characterized by permanent cell growth arrest in response to different stressors to avoid the propagation of genetically damaged cells [144]. Senescent cells do not transform into neoplastic cells because of their loss of growth capabilities. However, senescent cells change their gene expression profile and secrete multiple pro-inflammatory molecules (cytokines, chemokines, growth factors, and ptoteases) and extracellular vesicles, known as the senescence-associated secretory phenotype (SASP) [145]. There is substantial evidence that many factors from the SASP can create a microenvironment suitable for cancer development [146]. SASP has been shown to induce cancer promotion, progression, and metastasis in tumor cells from different tissues [147]. Thus, SASP may be one of the targets of cancer chemoprevention and treatment. Several attempts have been made to find phytochemicals or develop drugs that can target senescent cells (senotherapeutics) by interfering with their paracrine signaling (senomorphics) or selectively killing them (senolytics). These included metformin, rapamycin, 2,3-dihydroxybenzoic acid, simvastatin, quercetin, and fisetin. Recently, a phytochemical constituent found in a number of plants, rutin, has been found to have a remarkable capacity to target senescent cells by dampening the expression of the full spectrum SASP [148]. Different approaches, including the combination of a chemotherapeutic agent with rutin to target senescent cells, may be useful for cancer chemoprevention and treatment.

## 7. Clinical Trials of Chemoprevention Studies Using Phytochemicals and Limitations

As non-nutritive substances, phytochemicals are mainly classified into phenolics, carotenoids, organosulfur compounds, nitrogen-containing compounds, and alkaloids. They are known for their potential beneficial effects such as the prevention of various chronic diseases, including cancer [149]. Although many epidemiological studies suggest a significant association between the phytochemical consumption and a lower cancer risk of several types of cancer [149], these effects could not be reproduced in most clinical trials that were withdrawn early due to several reasons, including a lack of evidence or risk of side effects [149]. Therefore, human studies and/or clinical trials with safety assessments are still needed. There are limitations, including their low stability and bioavailability, that have hampered the application of phytochemicals in clinical trials [150]. To overcome the limitations, nanotechnology has attracted much attention due to its controlled release of phytochemicals, and thus the efforts may result in improving the pharmacokinetics and pharmacodynamics of phytochemicals.

## 8. Conclusions

In this short review, we introduce a combination of chemoprevention strategies as an approach targeting oncogenesis. Despite great advancements in prognosis, diagnosis, and treatment, cancer remains the second leading cause of disease-associated death worldwide. When driver gene(s) and actionable/druggable mutations are detected in cancer tissues by NGS, additional mutations may occur after using a molecular-targeted agent. Therefore, to fight cancer, the inhibition or retardation of the progression of preneoplastic lesions to malignant neoplasia during the early stages of oncogenesis is necessary. Therefore, chemopreventive strategies are important. When the chemopreventive effects of the administration of a single agent are limited owing to the low bioavailability and instability of agents, the combined use of different natural and synthetic compounds may be considered, as combined chemoprevention is expected to exert synergistic effects, reduce adverse effects, and possibly prevent drug resistance. Additionally, chemoprevention by targeting the SASP is promising because inflammation-associated oncogenesis requires a novel approach in order for us to fight it.

## Figures and Tables

**Figure 1 pharmaceuticals-16-01591-f001:**
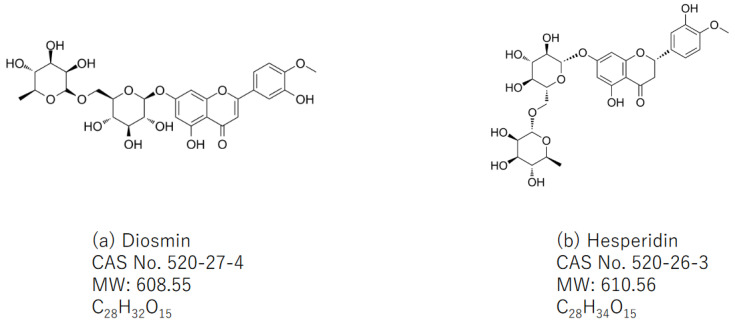
Chemical structures of (**a**) diosmin (CAS No. 520-27-4, MW: 608.55, C_28_H_32_O_15_) and (**b**) hesperidin (CAS No. 520-26-3, MW: 610.56, C_28_H_34_O_15_).

## Data Availability

Not applicable.

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
