# Peer review of "Potential Chemopreventive Effects of Dietary Combination of Phytochemicals against Cancer Development"

_pharmaceuticals, 2023, doi:10.3390/ph16111591_

Round 1

Reviewer 1 Report

Comments and Suggestions for Authors

A review study was conducted by the authors on the potential chemopreventive effects of combining two phytochemicals in dietary intake to prevent cancer development. They have summarized recent developments in combination chemoprevention and introduced their previous studies that prove the effectiveness of combining diosmin and hesperidin treatments. Both of these treatments have anti-inflammatory properties and were proven effective against chemically-induced carcinogenesis in several rodent tissues. In my opinion, the work can be accepted after the two minor revisions.

Minor points

1.           Improve Figure 1. Better resolution and eliminate nonpolar hydrogens.

2.           Add a paragraph regarding the molecular targets affected by diosmin and hesperidin.

Author Response

Reply to the comments by Reviewer 1

A review study was conducted by the authors on the potential chemopreventive effects of combining two phytochemicals in dietary intake to prevent cancer development. They have summarized recent developments in combination chemoprevention and introduced their previous studies that prove the effectiveness of combining diosmin and hesperidin treatments. Both of these treatments have anti-inflammatory properties and were proven effective against chemically-induced carcinogenesis in several rodent tissues. In my opinion, the work can be accepted after the two minor revisions.

Reply: Thank you very much for carefully reviewing our manuscript and for the valuable and positive comments for us to revise the manuscript.

Minor points

  1. Improve Figure 1. Better resolution and eliminate nonpolar hydrogens.

Reply: As suggested Figure 1 has been modified.

  1. Add a paragraph regarding the molecular targets affected by diosmin and hesperidin.

Reply: As recommended, the molecular targets of diosmin and hesperidin have been described with references by subheading “4.4. The molecular target of diosmin and hesperidin”.

Reviewer 2 Report

Comments and Suggestions for Authors

The presented review may be interesting, but in my opinion the authors did not use the huge potential of the topic at all. They cite very few valuable works, many citations are old and outdated, there are no tables or figures summarizing the available data, the authors cite a lot of their own research, which was published in the 1990s and is very outdated. Below I have presented various errors and inaccuracies. In its current form, the publication does not deserve publication because it does not bring anything new to the issue under discussion.

1. "Our research group previously reported the effective chemopreventive ability of several natural compounds" - such a statement should be in the body of the manuscript, not in the abstract part. Please remove it. 

2. lines 72-76 - "In this short review, we summarize recent advances in combination chemoprevention and introduce our previous studies that proved the efficacy of combination treatment with diosmin (Figure 1a) and hesperidin (Figure 1b), both of which have anti-inflammatory properties [18,34-39], against chemically-induced carcinogenesis in several rodent tissues [18,35,40-49]." - I do not understand what this fragment refers to and why the authors presented the structural formulas of diosmin and hesperidin below? The Authors are reviewing their own previous research? It is rather weird. Moreover the Authors cite their own research, which is rather old. 

3. lines 158-160 - "The most studied phytochemicals associated with oxidative damage are terpenoids, alkaloids, isothiocyanates, and polyphenols" - this sentence sounds incorrect to me, as someone could think that these phytochemicals are associated with oxidative damage and in fact it is completely opposite

4. lines 338-349 - there are various clinically used statins. Please specify which particular statins were used in the described experiments. 

Comments on the Quality of English Language

English requires minor correction. 

Author Response

Reply to the comments by Reviewer 2

The presented review may be interesting, but in my opinion the authors did not use the huge potential of the topic at all. They cite very few valuable works, many citations are old and outdated, there are no tables or figures summarizing the available data, the authors cite a lot of their own research, which was published in the 1990s and is very outdated. Below I have presented various errors and inaccuracies. In its current form, the publication does not deserve publication because it does not bring anything new to the issue under discussion.

Reply: Thank you very much for careful reviewing our manuscript. Your valuable comments are helpful for us to revise the manuscript.

  1. "Our research group previously reported the effective chemopreventive ability of several natural compounds" - such a statement should be in the body of the manuscript, not in the abstract part. Please remove it. 

Reply: I agree with you. As you suggested, the sentence has been deleted in the abstract,

  1. lines 72-76 - "In this short review, we summarize recent advances in combination chemoprevention and introduce our previous studies that proved the efficacy of combination treatment with diosmin (Figure 1a) and hesperidin (Figure 1b), both of which have anti-inflammatory properties [18,34-39], against chemically-induced carcinogenesis in several rodent tissues [18,35,40-49]." - I do not understand what this fragment refers to and why the authors presented the structural formulas of diosmin and hesperidin below? The Authors are reviewing their own previous research? It is rather weird. Moreover, the Authors cite their own research, which is rather old. 

Reply: I agree with you. The sentences have been moved to the last paragraph of the section entitled “4. Combined chemoprevention strategy”. Our studies are relatively old, but please note our challenging work.

  1. lines 158-160 - "The most studied phytochemicals associated with oxidative damage are terpenoids, alkaloids, isothiocyanates, and polyphenols" - this sentence sounds incorrect to me, as someone could think that these phytochemicals are associated with oxidative damage and in fact it is completely opposite.

Reply: I completely agree with you. The words “associated with oxidative damage” have been deleted.

  1. lines 338-349 - there are various clinically used statins. Please specify which particular statins were used in the described experiments. 

Reply: As suggested, we have added the generic names of statins used in the described experiments.

English requires minor corrections. 

Reply: The English of the revised text has been edited by a native speaker, Professor Brian Quinn, at Kyusyu University in Japan.

Reviewer 3 Report

Comments and Suggestions for Authors

Manuscript named Potential chemopreventive effects of dietary combination by phytochemicals against cancer development is interesting and comprehensive overview of current finding in phytotherapy for chemoprevention. 

Natural sources of the presented chemopreventive phytochemicals should be added in Section 4.

The following paragraph should be modified with generic names of the drugs:

"Combination drugs include coxib + aspirin [127], metformin + aspirin [128], coxib + EGFR tyrosine kinase inhibitor [129], statin + aspirin or xib [130], piroxicam + DFMO [131,132], NSAID + DFMO [133,134], and NSAID + 3-hydroxy-3-methylglutaryl-coenzyme A (HMG-CoA) reductase inhibitors [135,136]."

Comments on the Quality of English Language

Manuscript named Potential chemopreventive effects of dietary combination by phytochemicals against cancer development is well written. 

Author Response

Reply to the comments by Reviewer 3

Manuscript named Potential chemopreventive effects of dietary combination by phytochemicals against cancer development is interesting and comprehensive overview of current finding in phytotherapy for chemoprevention. 

Reply: Thank you very much for carefully reviewing our manuscript and for the valuable and positive comments for us to revise the manuscript.

Natural sources of the presented chemopreventive phytochemicals should be added in Section 4.

Reply: Thank you for your suggestion. As you recommended, we have added natural sources of diosmin and hesperidin by the subheading “4.1. The natural sources of diosmin and hesperidin”.

The following paragraph should be modified with generic names of the drugs:

"Combination drugs include coxib + aspirin [127], metformin + aspirin [128], coxib + EGFR tyrosine kinase inhibitor [129], statin + aspirin or xib [130], piroxicam + DFMO [131,132], NSAID + DFMO [133,134], and NSAID + 3-hydroxy-3-methylglutaryl-coenzyme A (HMG-CoA) reductase inhibitors [135,136]."

Reply: We have described the generic names of the drugs in the paragraph you suggested.

Comments on the Quality of English Language

Manuscript named Potential chemopreventive effects of dietary combination by phytochemicals against cancer development is well written. 

Reply: Many thanks for your positive comments on our manuscript.

Reviewer 4 Report

Comments and Suggestions for Authors

The review article by Tanaka, Aoki, and Tearasaki seeks to review the chemoprotective effects of dietary phytochemicals against cancer development. For the article to be accepted, it needs to be rewritten to be helpful to the cancer and herbal medicine community. There was no need to review the effective use of phytochemicals in randomized, clinical, or pre-clinical trials in humans. This is the most essential part that would be useful to review, given that in vitro, cellular, and animal tests are just initial stages. There are already numerous reviews of herbal medicines in in vitro trials. There is a need for a comprehensive review addressing the trials and effects studied in human clinical trials. Therefore, I suggest that the review article be redone and resubmitted.

Briefly, here are some ideas to improve the manuscript:

1. The authors should focus on the effective use of phytochemicals in randomized, clinical or pre-clinical trials in humans rather than just in vitro, cellular and animal tests. A comprehensive review addressing the trials and effects studied in human clinical trials would benefit the cancer and herbal medicine community more.

2. The authors should provide more specific examples of dietary phytochemicals showing chemoprotective effects against cancer development. This would help the readers better understand the effectiveness of these compounds.

3. The authors should highlight the potential limitations of using dietary phytochemicals as chemoprotective agents, such as the optimal dose, the duration of treatment, and the possible interactions with other drugs. This would help the readers make informed decisions about using these compounds.

Comments on the Quality of English Language

English should be improved. 

Author Response

Reply to the comments by Reviewer 4

The review article by Tanaka, Aoki, and Tearasaki seeks to review the chemoprotective effects of dietary phytochemicals against cancer development. For the article to be accepted, it needs to be rewritten to be helpful to the cancer and herbal medicine community. There was no need to review the effective use of phytochemicals in randomized, clinical, or pre-clinical trials in humans. This is the most essential part that would be useful to review, given that in vitro, cellular, and animal tests are just initial stages. There are already numerous reviews of herbal medicines in in vitro trials. There is a need for a comprehensive review addressing the trials and effects studied in human clinical trials. Therefore, I suggest that the review article be redone and resubmitted.

Reply: Thank you very much for carefully reviewing our manuscript and for the valuable comments for us to revise the manuscript.

Briefly, here are some ideas to improve the manuscript:

  1. The authors should focus on the effective use of phytochemicals in randomized, clinical or pre-clinical trials in humans rather than just in vitro, cellular and animal tests. A comprehensive review addressing the trials and effects studied in human clinical trials would benefit the cancer and herbal medicine community more.

Reply: We have added the chemoprevention clinical trials using phytochemicals that are not easy to perform because of low bioavailability and side effects. This has been described in the section entitled “6. Clinical trials of chemoprevention studies by phytochemicals and limitations”.

  1. The authors should provide more specific examples of dietary phytochemicals showing chemoprotective effects against cancer development. This would help the readers better understand the effectiveness of these compounds.

Reply: As suggested, representative potential phytochemicals with chemopreventive ability are listed in the last paragraph of the section entitled “Phytochemicals”. We also have briefly documented specific examples by citing a reference (No. 149) which is an elegant and latest review article regarding epidemiological data and clinical trials of cancer prevention by phytochemicals.

  1. The authors should highlight the potential limitations of using dietary phytochemicals as chemoprotective agents, such as the optimal dose, the duration of treatment, and the possible interactions with other drugs. This would help the readers make informed decisions about using these compounds.

Reply: In the new section entitled “6. Clinical trials of chemoprevention studies by phytochemicals and limitations”, we have added the potential limitations of using dietary phytochemicals as chemoprotective agents, as you indicated.

English should be improved. 

Reply: The English of the revised text has been edited by a native speaker, Professor Brian Quinn, at Kyusyu University in Japan.

Round 2

Reviewer 2 Report

Comments and Suggestions for Authors

I still do not understand what is the purpose of such a review. In the title there is "... dietary combination by phytochemicals against cancer development" - what dietary combination did Author mention in the text? Did they refer to any studies related to dietary intake of any non-nutritive compunds where any cancr prevention was proved? After the first round of review the Authors have added Paragraph No 6 "6. Clinical trials of chemoprevention studies by phytochemicals and limiations", in which they wrote "Although many epidemiological studies suggest a significant association between the phytochemical consumption and a lower cancer risk of several types of cancer [149], these effects could not be reproduced in the most clinical trials that were withdrawn early due to several reasons, including a lack of evidence or risk of side effects [149]. " So, no human study have confirmed the effectiveness of non-nutritive compund in cancer prevention? 

I do not see any detailed infromation on any phytochemicals, except hesperidin and diosmin, for which the Authors have provided their own research results (which are rather old). Therefore I still do not see any reasons to publish such a review. 

Comments on the Quality of English Language

Dear Editor,

the manuscript was very little revised. I still do not see any reason to publish such a review paper. I will not change my decision, so if you want to push this paper forward, please choose another reviewer.

Best regards